# A tempered fractional Hawkes framework for finite-memory drought dynamics

Mauricio Herrera-Marín

Faculty of Engineering, Universidad del Desarrollo, Santiago, Chile Avda. Plaza 700, San Carlos de Apoquindo, Las Condes. Santiago, Chile.

**Correspondence:** Mauricio Herrera-Marín (mherrera@udd.cl)

Abstract. Meteorological droughts emerge from nonlinear land-atmosphere feedbacks and circulation anomalies, whose persistence and recurrence are not well represented by conventional stochastic models with exponentially decaying memory. Here we introduce a *Tempered Fractional Hawkes Process* (TFHP) to describe drought onsets as a self-exciting point process with algebraically decaying but ultimately finite memory. In this formulation, the conditional intensity of dry-spell initiation obeys a tempered fractional differential equation in the Caputo sense, where the kernel  $\phi(t) \propto t^{-\alpha}e^{-\theta t}$  combines long-range dependence with exponential tempering that enforces dynamic stability. The model parameters have clear physical meaning:  $\mu$  (baseline exogenous activity),  $\kappa$  (self-excitation strength),  $\alpha$  (fractional memory order), and  $\theta^{-1}$  (finite-memory horizon). Using 43 years of daily ERA5 precipitation over continental Chile, we estimate spatial fields of  $(\mu,\kappa,\alpha)$  and derive an effective memory timescale  $\tau_m=1/\theta_{\rm eff}$ . Results reveal geographically organised persistence regimes: subtropical arid and high-latitude subpolar regions exhibit slowly decaying memory and strong endogenous reinforcement, whereas mid-latitude zones display faster relaxation and weaker feedbacks. The TFHP thus offers a parsimonious and physically interpretable representation of finite climatic memory, bridging fractional calculus, point-process theory, and nonlinear geophysical dynamics. Beyond drought analysis, it provides a general framework to quantify persistence, clustering, and resilience in non-Markovian environmental systems.

#### 15 1 Introduction

Meteorological droughts are among the most complex manifestations of hydroclimatic variability, emerging from nonlinear interactions among precipitation, soil moisture, and large-scale circulation patterns. Their onset, persistence, and termination are modulated by feedbacks acting across a wide range of temporal and spatial scales—from synoptic blocking and subtropical anticyclones to land—atmosphere coupling and oceanic teleconnections (Seneviratne et al., 2010; Miralles et al., 2019; Garreaud et al., 2017; Bozkurt et al., 2018; Boers et al., 2023). Quantifying these multiscale processes and their associated persistence remains a central challenge in hydroclimatology, with direct implications for water-resource management, ecosystem resilience, and climate adaptation. In particular, the question of *how memory operates* within the coupled land—atmosphere system, and how it modulates the recurrence of dry spells, lies at the heart of modern drought research (Vogel et al., 2021; Raymond et al., 2022).

Traditional stochastic formulations of drought occurrence—such as Poisson, Markov chain, and renewal processes (Wilks, 1999; Srikanthan and McMahon, 2017)—assume either independence or exponentially decaying correlations between events. Although computationally tractable, these frameworks cannot reproduce the long-range dependence, clustering, and heavy-tailed duration distributions that characterize observed dry-spell statistics (Franzke, 2012; Serinaldi and Kilsby, 2017; Marani and Zorzetto, 2022; Katul and Porporato, 2007).

They also neglect the nonlinear feedbacks between antecedent and subsequent conditions, which play a central role in sustaining drought persistence through soil moisture–atmosphere coupling and radiative effects (Seneviratne et al., 2010; Miralles et al., 2019; Boers et al., 2023). As a result, classical approaches underestimate both the probability of prolonged droughts and the internal organization of their temporal structure (Livneh et al., 2023).

Self-exciting point processes provide an alternative stochastic description that explicitly incorporates internal feedback. The Hawkes process (Hawkes, 1971; Bacry et al., 2015) has been widely applied to model clustered events whose occurrence rate temporarily increases after previous occurrences, capturing the self-reinforcing nature of complex systems. In hydroclimatology, Hawkes-type models have recently been used to characterize the temporal clustering of rainfall extremes, heatwaves, and drought onsets (Cheng et al., 2014; Baran et al., 2022; Zhang et al., 2023). By decomposing the conditional intensity  $\lambda(t)$  into exogenous and endogenous components, these models distinguish between spontaneous background activity ( $\mu$ ) and self-excitation ( $\kappa$ ), providing a natural measure of internal feedback strength. However, the standard exponential kernel  $\phi(t) = \alpha \beta e^{-\beta t}$  enforces a strictly Markovian relaxation, with exponentially decaying correlations that are unable to reproduce the slow, scale-free persistence found in hydrometeorological extremes (Barabási, 2005; Franzke, 2012; Fan et al., 2023). Consequently, conventional Hawkes formulations fail to represent the heavy-tailed waiting times and long-memory clustering typical of hydroclimatic processes.

Fractional calculus offers a rigorous mathematical extension for representing long-memory and nonlocal dynamics (Podlubny, 1998; Kilbas et al., 2006; Mainardi, 2010). Fractional derivatives generalize differentiation by integrating the full history of a process, thereby producing algebraically decaying memory consistent with the persistence observed in climate and hydrology (Meerschaert and Sikorskii, 2012; Chechkin and Metzler, 2017). Among the various formulations, the Caputo fractional derivative provides a physically interpretable causal operator that preserves standard initial conditions and has found increasing use in geosciences (Hilfer, 2000; Gorenflo et al., 2014; Chen et al., 2021). Fractional operators have been successfully applied to model relaxation, diffusion, and scaling behavior in nonlinear geophysical systems (Jiao et al., 2021; Tarasov, 2019), bridging stochastic dynamics, anomalous transport, and climatic variability. Recent studies in Nonlinear Processes in Geophysics have demonstrated the usefulness of fractional frameworks for describing persistence and scaling laws in precipitation, temperature, and river discharge records (Serinaldi and Kilsby, 2017; Grigoriu, 2021; Marani and Zorzetto, 2022), motivating their integration with event-based point-process representations.

Building upon these theoretical and empirical advances, we introduce a *Tempered Fractional Hawkes Process (TFHP)* to model drought onsets as a self-exciting process with algebraically decaying yet finite memory. The TFHP extends the fractional Hawkes formulation (Hainaut, 2020; Habyarimana et al., 2023) by incorporating an exponential tempering factor

that regularizes the power-law memory kernel:

60 
$$\phi_{\theta}(t) = \frac{\kappa e^{-\theta t} t^{\alpha - 1}}{\Gamma(\alpha)},$$

where  $\alpha$  controls the rate of memory decay,  $\kappa$  quantifies self-excitation strength, and  $\theta$  defines the characteristic damping timescale. The corresponding mean-field dynamics are governed by a *tempered Caputo fractional differential equation*,

$$D_{C,\theta}^{\alpha}\lambda(t) = \kappa\lambda(t) + \frac{\mu\,e^{-\theta t}\,t^{-\alpha}}{\Gamma(1-\alpha)},$$

which reduces to the classical Hawkes process as  $\alpha \to 1^-$  and  $\theta \to \infty$ , and to the pure fractional regime as  $\theta \to 0^+$ . Exponential tempering introduces a finite-memory horizon  $\tau_m = 1/\theta$ , ensuring dynamically stable and physically interpretable behavior while preserving algebraic persistence at intermediate scales (Meerschaert and Sabzikar, 2015; Zhou et al., 2021; Li et al., 2022). This mathematical regularization captures a key physical property of hydroclimatic systems: although droughts exhibit long-range dependence, they are ultimately bounded by finite soil moisture storage and seasonal atmospheric reinitialization.

Conceptually, the TFHP bridges two complementary perspectives of drought dynamics: (1) the discrete-event representation of drought initiation through self-exciting point processes, and (2) the continuous-time relaxation of climatic memory described by fractional operators. The fractional order  $\alpha$  quantifies the strength of long-range persistence, while the tempering parameter  $\theta$  limits memory depth, preventing unbounded accumulation of past influence. Together,  $(\mu, \kappa, \alpha, \theta)$  define a parsimonious but physically meaningful model of drought occurrence that links stochastic event clustering to hydroclimatic feedback mechanisms. By situating fractional dynamics within the Hawkes formalism, the TFHP provides an interpretable bridge between stochastic process theory, nonlinear climate dynamics, and resilience analysis.

Using 43 years of daily ERA5 precipitation data across continental Chile, we estimate the spatial distribution of  $(\mu, \kappa, \alpha)$  and infer the corresponding effective finite-memory timescale  $\tau_m = 1/\theta_{\rm eff}$ . This analysis reveals coherent spatial patterns of drought persistence, excitation strength, and background hazard, allowing a process-based quantification of climatic memory across contrasting hydroclimatic regimes—from hyper-arid subtropical zones to temperate mid-latitudes and humid southern regions. The Chilean domain offers a natural climatic transect that spans multiple regimes of drought persistence, making it an ideal testbed for evaluating how fractional memory and self-excitation interact under diverse atmospheric forcing conditions (Bozkurt et al., 2018; Garreaud et al., 2017, 2020).

The remainder of this article is organized as follows. Section 2 presents the mathematical formulation of the TFHP and its equivalence with a tempered fractional differential equation. Section 3 describes the dataset, event extraction, and parameter estimation methodology. Section 4 reports the spatial organization of the inferred parameters and their climatic interpretation. Finally, Section 5 discusses theoretical implications, methodological advances, and possible extensions toward fractional—tempered models for climate predictability and resilience diagnostics.

The objectives of this study are threefold. First, we formulate a tempered fractional Hawkes framework that integrates self-excitation, long-range dependence, and finite-memory damping into a unified stochastic-dynamical model of drought onset. Second, we develop a practical inference scheme to estimate  $(\mu, \kappa, \alpha)$  from observed dry-spell sequences and to derive the effective memory horizon  $\tau_m = 1/\theta_{\rm eff}$ . Finally, we apply this framework to four decades of ERA5 precipitation across Chile

110

to identify spatial regimes of memory and feedback. Beyond its regional application, the proposed framework contributes to a broader understanding of persistence in hydroclimatic extremes by demonstrating that drought dynamics can be described as a *tempered fractional process*: a system that retains scale-free memory yet remains dynamically stable due to finite physical constraints. This formulation opens a pathway for integrating memory-aware stochastic processes into climate modeling, risk assessment, and nonlinear predictability studies, providing a physically interpretable diagnostic of climatic resilience across timescales.

#### 2 Mathematical Framework

## 2.1 Self-exciting drought onset model as a point process

Let  $\{N(t), t \geq 0\}$  denote a simple counting process representing the occurrence of drought onset events (dry-spell initiation) at times  $\{t_i\}$ , and let  $\mathcal{H}_t = \{t_i : t_i < t\}$  denote the event history up to time t. We assume that N(t) is conditionally Poisson with *conditional intensity*  $\lambda(t|\mathcal{H}_t)$  such that

$$\mathbb{P}\{N(t+\mathrm{d}t) - N(t) = 1 \mid \mathcal{H}_t\} = \lambda(t|\mathcal{H}_t)\,\mathrm{d}t + o(\mathrm{d}t). \tag{1}$$

In the Hawkes framework, the conditional intensity is written as

105 
$$\lambda(t|\mathcal{H}_t) = \mu + \int_0^t \phi(t-s) \, \mathrm{d}N(s), \tag{2}$$

where  $\mu > 0$  represents the exogenous (background) rate, and  $\phi(t)$  is a causal, nonnegative memory kernel that quantifies self-excitation: each past drought onset at time s increases the instantaneous hazard of a new onset at time t > s.

Classical Hawkes models adopt an exponential kernel  $\phi(t) = \alpha \beta e^{-\beta t}$ , which enforces Markovian memory with rapid exponential decay (Hawkes, 1971; Bacry et al., 2015). Such kernels cannot reproduce the slowly decaying, power-law correlations and heavy-tailed waiting times observed in hydroclimatic drought sequences (Franzke, 2012; Palma and Letelier, 2020). To capture long-range dependence, we replace  $\phi$  with a fractional power-law kernel.

## 2.2 Fractional Hawkes kernel and mean-field dynamics

We first introduce the fractional Hawkes process, in which

$$\phi_{\alpha}(t) = \frac{\kappa t^{\alpha - 1}}{\Gamma(\alpha)}, \qquad 0 < \alpha < 1, \quad \kappa > 0, \tag{3}$$

where  $\alpha$  is the fractional memory order and  $\Gamma(\cdot)$  is the Gamma function. This kernel decays algebraically as  $t^{\alpha-1}$ , implying non-Markovian memory with a power-law tail. Small  $\alpha$  corresponds to extremely persistent excitation, i.e. slow memory decay. Taking conditional expectation of (2) with respect to  $\mathcal{H}_t$  yields the *mean-field intensity*,

$$\lambda(t) = \mu + \int_{0}^{t} \phi_{\alpha}(t-s)\lambda(s) \,\mathrm{d}s. \tag{4}$$

Substituting (3) gives

120 
$$\lambda(t) = \mu + \frac{\kappa}{\Gamma(\alpha)} \int_{0}^{t} (t - s)^{\alpha - 1} \lambda(s) \, \mathrm{d}s = \mu + \kappa I^{\alpha}[\lambda](t), \tag{5}$$

where  $I^{\alpha}$  denotes the Riemann–Liouville fractional integral of order  $\alpha$ ,

$$I^{\alpha}[f](t) = \frac{1}{\Gamma(\alpha)} \int_{0}^{t} (t-s)^{\alpha-1} f(s) \, \mathrm{d}s. \tag{6}$$

Equation (5) is a nonlinear Volterra equation for  $\lambda(t)$ , and it can be recast as a fractional differential equation using Caputo calculus.

## 125 2.3 Caputo formulation and fractional relaxation

Applying the Caputo fractional derivative  $D_C^{\alpha}$  to (5) and recalling the identity  $D_C^{\alpha}I^{\alpha}f(t)=f(t)$  for sufficiently smooth f, we obtain

$$D_C^{\alpha}\lambda(t) = D_C^{\alpha}\mu + \kappa\lambda(t),\tag{7}$$

with

130 
$$D_C^{\alpha} \mu = \frac{\mu t^{-\alpha}}{\Gamma(1-\alpha)}.$$
 (8)

Therefore, the mean-field intensity of the fractional Hawkes process satisfies

$$D_C^{\alpha}\lambda(t) = \kappa\lambda(t) + \frac{\mu t^{-\alpha}}{\Gamma(1-\alpha)}.$$
(9)

For sufficiently large t, the inhomogeneous forcing term  $\mu t^{-\alpha}/\Gamma(1-\alpha)$  decays algebraically, and the asymptotic dynamics approach

135 
$$D_C^{\alpha}\lambda(t) \simeq \kappa \lambda(t) + \varepsilon(t),$$
 (10)

where  $\varepsilon(t)$  collects residual exogenous fluctuations. Equation (9) is a fractional relaxation law: the current drought hazard  $\lambda(t)$  relaxes according to a Caputo derivative of order  $\alpha$ , indicating that the system retains an algebraically weighted memory of its entire past. In the limit  $\alpha \to 1^-$ ,  $D_C^{\alpha}\lambda(t) \to \mathrm{d}\lambda/\mathrm{d}t$  and (9) reduces to the classical first-order relaxation  $\dot{\lambda}(t) = \kappa\lambda(t) + \mu$ .

# 2.4 Analytical solution: Mittag-Leffler response

Taking the Laplace transforms  $\mathcal{L}\{f(t)\} = \tilde{f}(s)$  and using  $\mathcal{L}\{D_C^{\alpha}f\} = s^{\alpha}\tilde{f}(s) - s^{\alpha-1}f(0)$ , Eq. (9) gives

$$s^{\alpha}\tilde{\lambda}(s) - s^{\alpha - 1}\lambda(0) = \kappa\tilde{\lambda}(s) + \frac{\mu\Gamma(1 - \alpha)}{\Gamma(1 - \alpha)}s^{\alpha - 1} + \tilde{\varepsilon}(s),\tag{11}$$

so that

$$\tilde{\lambda}(s) = \frac{s^{\alpha - 1}\lambda(0) + \tilde{\varepsilon}(s)}{s^{\alpha} - \kappa} + \frac{\mu s^{\alpha - 1}}{s^{\alpha} - \kappa}.$$
(12)

Inverting the Laplace transform yields the well-known Mittag-Leffler relaxation (Mainardi, 2010; Hainaut, 2020; Chen et al., 2021):

$$\lambda(t) = \lambda(0) E_{\alpha}(\kappa t^{\alpha}) + \mu E_{\alpha}(\kappa t^{\alpha})$$

$$+ \int_{0}^{t} (t - s)^{\alpha - 1} E_{\alpha, \alpha} [\kappa(t - s)^{\alpha}] \varepsilon(s) \, \mathrm{d}s,$$
(13)

where  $E_{\alpha}(z)$  and  $E_{\alpha,\beta}(z)$  denote the one- and two-parameter Mittag–Leffler functions,

$$E_{\alpha}(z) = \sum_{n=0}^{\infty} \frac{z^n}{\Gamma(\alpha n + 1)}, \qquad E_{\alpha,\beta}(z) = \sum_{n=0}^{\infty} \frac{z^n}{\Gamma(\alpha n + \beta)}.$$
 (14)

These functions generalize the exponential: for  $\alpha = 1$ ,  $E_1(z) = e^z$ . For  $0 < \alpha < 1$ ,  $E_{\alpha}(\kappa t^{\alpha})$  decays as a power law rather than an exponential, implying extremely persistent memory. This reproduces the empirically observed clustering of dry spells and long drought persistence in hydroclimatic records (Franzke, 2012; Palma and Letelier, 2020).

# 2.5 Tempered fractional Hawkes kernel: finite-memory regularization

Pure power-law kernels imply *infinite memory*: all past events, no matter how old, continue to influence  $\lambda(t)$  with nonzero weight. This is mathematically consistent but physically problematic for hydroclimate, where land-atmosphere coupling, soil storage, and synoptic circulation impose a finite persistence horizon.

To regularize the memory, we introduce an exponential tempering factor  $e^{-\theta t}$  into the kernel:

$$\phi_{\alpha,\theta}(t) = \frac{\kappa e^{-\theta t} t^{\alpha - 1}}{\Gamma(\alpha)}, \qquad 0 < \alpha < 1, \quad \kappa > 0, \quad \theta > 0.$$
(15)

The corresponding mean-field equation is now

$$\lambda(t) = \mu + \int_{0}^{t} \phi_{\alpha,\theta}(t-s) \,\lambda(s) \,\mathrm{d}s = \mu + \kappa I_{\theta}^{\alpha}[\lambda](t),\tag{16}$$

160 where

$$I_{\theta}^{\alpha}[f](t) = \frac{1}{\Gamma(\alpha)} \int_{0}^{t} e^{-\theta(t-s)} (t-s)^{\alpha-1} f(s) \, \mathrm{d}s \tag{17}$$

is a tempered fractional integral (Meerschaert and Sabzikar, 2015; Li et al., 2022). For  $\theta \to 0^+$ ,  $I_{\theta}^{\alpha} \to I^{\alpha}$  and (16) reduces to (5). For  $\theta > 0$ , the memory kernel is algebraic at short lags but exponentially suppressed at long lags, introducing a finite effective memory horizon  $\tau_m = 1/\theta$ .

## 165 2.6 Tempered Caputo operator and finite-memory dynamics

The tempered counterpart of the Caputo derivative is defined as (Meerschaert and Sabzikar, 2015)

$$D_{C,\theta}^{\alpha}f(t) = e^{-\theta t}D_{C}^{\alpha}\left[e^{\theta t}f(t)\right]$$

$$= \frac{1}{\Gamma(1-\alpha)} \int_{0}^{t} e^{-\theta(t-s)}(t-s)^{-\alpha}f'(s) \,\mathrm{d}s. \tag{18}$$

For sufficiently smooth f, one has the inversion identity

$$D_{C\theta}^{\alpha} I_{\theta}^{\alpha} f(t) = f(t), \tag{19}$$

70 which generalizes  $D_C^{\alpha} I^{\alpha} f(t) = f(t)$ .

Applying  $D_{C}^{\alpha}$  to (16) yields the tempered fractional Hawkes equation

$$D_{C,\theta}^{\alpha}\lambda(t) = \kappa\lambda(t) + \frac{\mu e^{-\theta t} t^{-\alpha}}{\Gamma(1-\alpha)}.$$
(20)

Equation (20) reduces to the untempered form (9) in the limit  $\theta \to 0^+$ . In the opposite limit  $\alpha \to 1^-$  and large  $\theta$ ,  $D_{C,\theta}^{\alpha}$  approaches the standard first derivative plus an exponential damping term, recovering approximately Markovian relaxation. Thus, Eq. (20) interpolates continuously between: (i) long-memory, power-law persistence (fractional,  $\theta \approx 0$ ), and (ii) finite-memory, exponentially truncated persistence (tempered,  $\theta > 0$ ).

## 2.7 Analytical structure of the tempered solution

Taking Laplace transforms of (20) leads to

$$\tilde{\lambda}(s) = \frac{s^{\alpha - 1}\lambda(0) + \tilde{\varepsilon}(s)}{(s + \theta)^{\alpha} - \kappa} + \frac{\mu s^{\alpha - 1}}{(s + \theta)^{\alpha} - \kappa},\tag{21}$$

which can be inverted in terms of tempered Mittag-Leffler functions  $E_{\alpha}^{(\theta)}$  and  $E_{\alpha,\beta}^{(\theta)}$  (Zhou et al., 2021; Li et al., 2022). These functions behave like Mittag-Leffler functions at short times (algebraic memory) and acquire an exponential cutoff at long times. Consequently,  $\lambda(t)$  exhibits finite relaxation toward  $\mu$  on a characteristic timescale  $\tau_m = 1/\theta$ , eliminating the physically unrealistic "infinite memory" of the pure power-law kernel.

# 2.8 Numerical inference of $(\mu, \kappa, \alpha)$

For each station, we treat the observed sequence of drought onsets  $\{t_i\}$  as a realization of a Hawkes process with kernel (3). We estimate the parameters  $(\mu, \kappa, \alpha)$  by maximizing the log-likelihood of the (untempered) fractional Hawkes model. Denote by  $\lambda(t|\mathcal{H}_t; \mu, \kappa, \alpha)$  the conditional intensity constructed via (2)–(3). The log-likelihood over an observation window [0, T] is

$$\mathcal{L}(\mu, \kappa, \alpha) = \sum_{t_i \le T} \log \lambda(t_i | \mathcal{H}_{t_i}; \mu, \kappa, \alpha) - \int_0^T \lambda(t | \mathcal{H}_t; \mu, \kappa, \alpha) \, \mathrm{d}t.$$
 (22)

The first term rewards the model for assigning high intensity to observed onset times; the second penalizes over-predicting events. To evaluate  $\lambda(t|\mathcal{H}_t)$  efficiently on a dense time grid, we compute the convolution

$$(\phi_{\alpha} * N)(t) = \int_{0}^{t} \frac{\kappa (t - s)^{\alpha - 1}}{\Gamma(\alpha)} \, \mathrm{d}N(s)$$

via FFT-based methods (on GPU when available), or via Numba-optimized causal accumulation on CPU. We then optimize (22) using a hybrid global–local strategy (e.g., dual annealing followed by L-BFGS-B) under physically motivated constraints  $\mu > 0$ ,  $0 < \alpha < 1$ ,  $\kappa > 0$ , ensuring subcriticality.

This yields spatial fields of  $\mu$  (baseline drought hazard),  $\kappa$  (self-excitation strength), and  $\alpha$  (fractional memory order) for each station.

#### 2.9 Effective tempering scale and memory horizon

Directly fitting  $(\mu, \kappa, \alpha, \theta)$  for the full tempered kernel (15) at each station is possible in principle via a four-parameter likelihood, but is numerically fragile for short or weakly clustered records. Instead, we define an *effective tempering rate*  $\theta_{\text{eff}}$  200 by enforcing that the total endogeneity of the tempered kernel matches a prescribed subcriticality level  $\eta^*$ , common across stations.

For a Hawkes process with kernel  $\phi(t)$ , the integrated endogeneity (branching ratio) is

$$\eta = \int_{0}^{\infty} \phi(t) \, \mathrm{d}t.$$

For the fractional (untempered) kernel (3), one obtains

$$\eta_{\text{frac}} = \frac{\kappa}{\Gamma(1+\alpha)}$$
, (23)

which measures how strongly events trigger subsequent events. For the tempered kernel (15), integration gives

$$\eta_{\text{temp}} \approx \frac{\kappa}{\theta^{\alpha}}.$$
(24)

We define  $\theta_{\mathrm{eff}}$  for each station by solving

$$\eta_{\text{temp}}(\mu, \kappa, \alpha, \theta_{\text{eff}}) = \eta^*,$$
(25)

for a fixed  $\eta^*

interpreted as the characteristic time over which drought history continues to elevate the local hazard before relaxing toward the baseline rate  $\mu$  under a common, near-critical excitation budget  $\eta^*$ .

Equations (26)–(27) provide a robust way to map spatial variations in drought persistence even when direct four-parameter estimation  $(\mu, \kappa, \alpha, \theta)$  is statistically unstable. Importantly,  $\tau_m$  is not an arbitrary fitting constant: it is a diagnostic of *how much memory the system would need to retain in order to remain subcritical while reproducing its observed self-excitation strength*  $\kappa$  and memory order  $\alpha$ . Regions with large  $\tau_m$  behave as high-memory systems, in which antecedent dry spells continue to influence future drought onset risk over extended periods, whereas regions with small  $\tau_m$  relax rapidly and exhibit weak event-to-event persistence.

In summary, the Tempered Fractional Hawkes Process (TFHP) links physically interpretable drought dynamics to a mathematically rigorous fractional-tempered relaxation law. The parameters  $(\mu, \kappa, \alpha)$  are inferred directly from observed drought onset sequences via likelihood maximization of the fractional Hawkes model, and the derived scale  $\tau_m = 1/\theta_{\rm eff}$  quantifies an effective finite-memory horizon. This establishes a bridge between statistical event clustering and hydroclimatic memory, enabling spatial diagnostics of drought persistence and resilience.

#### 3 Data and Methods

# 3.1 Data sources and spatial domain

Daily precipitation data were obtained from the ERA5 reanalysis (Hersbach et al., 2023), which provides global coverage at  $0.25^{\circ} \times 0.25^{\circ}$  spatial resolution. The study domain encompasses continental Chile, extending from  $17.5^{\circ}$ S to  $56.0^{\circ}$ S and from  $76.0^{\circ}$ W to  $66.0^{\circ}$ W, thereby capturing the country's full climatic gradient—from hyper-arid deserts in the north to humid subpolar regions in the south. Although national agencies provide in-situ rainfall records (CR2, 2024), many stations contain gaps or short time spans (

255

onsets separated by fewer than two days were merged. The event times were expressed in continuous days from the start of the record and normalized to a common interval [0,T] for numerical stability and comparability across sites.

#### 3.3 Numerical discretization of the fractional Hawkes intensity

The fractional Hawkes intensity (Eq. (5)) involves the convolution between the event series N(t) and the power-law memory kernel  $(t-s)^{\alpha-1}/\Gamma(\alpha)$ ,

$$(\phi_{\alpha} * N)(t) = \int_{0}^{t} \frac{\kappa(t-s)^{\alpha-1}}{\Gamma(\alpha)} \, \mathrm{d}N(s).$$

250 For discrete event sequences, this integral becomes a sum over all previous onsets. To evaluate it efficiently for hundreds of time series, a hybrid numerical strategy was employed. Long records were computed using FFT-based convolution,

$$\lambda(t_k) = \mu + \mathcal{F}^{-1}[\mathcal{F}[\phi_\alpha] \cdot \mathcal{F}[N]](t_k),$$

reducing computational cost from  $O(N^2)$  to  $O(N\log N)$ , while sparse sequences were handled through optimized causal accumulation implemented with Numba. Kernel truncation at a finite lag  $L_{\rm max}$  ensured numerical stability, with  $(L_{\rm max})^{\alpha-1}/\Gamma(\alpha) < 10^{-6}$  introducing negligible error for  $\alpha > 0.1$ . This discretized intensity  $\lambda(t)$  provides the basis for likelihood estimation.

#### 3.4 Parameter estimation via maximum likelihood

The parameters  $(\mu, \kappa, \alpha)$  were inferred independently for each spatial location by maximizing the log-likelihood

$$\mathcal{L}(\mu, \kappa, \alpha) = \sum_{i} \log \lambda(t_i | \mathcal{H}_{t_i}) - \int_{0}^{T} \lambda(t | \mathcal{H}_{t}) dt,$$

subject to  $\mu > 10^{-5}$ ,  $0.05 < \alpha < 0.95$ , and  $\kappa > 10^{-5}$ . Optimization proceeded in two stages: a global search using the dual-annealing algorithm, followed by local refinement with L-BFGS-B using analytical gradients. The integral term was approximated by trapezoidal quadrature over the discretized  $\lambda(t)$ . To improve convergence and comparability, all time series were rescaled to a normalized temporal domain [0,100] prior to fitting. This two-step maximum-likelihood procedure yields stable and interpretable estimates even for heterogeneous event densities across Chile's climatic regions.

## 3.5 Estimation of the effective tempering rate and memory horizon

Direct joint estimation of  $(\mu, \kappa, \alpha, \theta)$  from the tempered kernel (Eq. (15)) can be unstable when event counts are limited or parameters are highly correlated. Therefore, an *effective tempering rate*  $\theta_{\text{eff}}$  was derived from the subcriticality condition  $\eta_{\text{temp}}(\theta) = \kappa/\theta^{\alpha} = \eta^*$ , where  $\eta^* = 0.9$  denotes a prescribed endogeneity level ensuring dynamic stability. Solving for  $\theta_{\text{eff}}$  gives

$$\theta_{\mathrm{eff}} = \left(\frac{\kappa}{\eta^*}\right)^{1/\alpha}, \qquad \tau_m = 1/\theta_{\mathrm{eff}},$$

where  $\tau_m$  defines the effective finite-memory horizon. This diagnostic quantifies the minimal damping required for stability while retaining the observed excitation and persistence characteristics. Large  $\tau_m$  values indicate long-memory regimes where past droughts exert extended influence, whereas small  $\tau_m$  corresponds to short-memory, rapidly relaxing dynamics.

# 3.6 Implementation and validation

All computations were performed in Python 3.9 using NumPy, SciPy, CuPy, and Numba for numerical acceleration, and Matplotlib and Cartopy for visualization. Parameter estimation was parallelized across grid points to process the 460-site network efficiently on multi-core and GPU platforms. Robustness was assessed through Monte Carlo perturbations in which 5% of onset events were randomly removed; mean deviations remained below 5% for  $\mu$ , 8% for  $\kappa$ , and 10% for  $\alpha$ , confirming parameter stability. Model adequacy was evaluated by comparing empirical and model-based waiting-time distributions and by examining the uniformity of the residual process  $u_i = 1 - \exp[-\Lambda(t_i)]$ , where  $\Lambda(t) = \int_0^t \lambda(s) \, ds$  denotes the cumulative intensity. These diagnostics confirmed that the fractional Hawkes model accurately reproduces both clustering and heavy-tailed inter-onset intervals observed in the data, outperforming exponential (Markovian) Hawkes and renewal process formulations across all hydroclimatic regimes.

#### 4 Results

#### 4.1 Spatial distribution of Hawkes parameters

Figure 1 shows the spatial distributions of the inferred parameters of the fractional Hawkes process: the baseline rate  $\mu$  (panel A), the fractional memory order  $\alpha$  (panel B), and the excitation amplitude  $\kappa$  (panel C). Together, these parameters decompose drought onset dynamics into: (i) the externally driven occurrence rate ( $\mu$ ), (ii) the endogenous reinforcement or feedback strength ( $\kappa$ ), and (iii) the persistence of past influence ( $\alpha$ ).

The baseline rate  $\mu$  exhibits a clear meridional gradient (Fig. 1A), increasing toward lower latitudes. Values of  $\mu > 0.6$  in northern and north-central Chile indicate a high background frequency of dry days, consistent with arid and semi-arid regimes. In contrast, the humid south is characterized by  $\mu 

305

in the arid north, weak soil-moisture buffering and recurring circulation blocking can maintain deficits for extended periods, whereas the humid south experiences more frequent frontal passage and hydrological resetting.

The excitation amplitude  $\kappa$  (Fig. 1C) reveals localized maxima (values of 2–5) in the north-central corridor and parts of central Chile, indicating strong endogenous reinforcement of dry-spell initiation. In these regions, once dryness is established, the conditional probability of additional drought onsets is substantially elevated, suggesting a self-sustaining regime shaped by land–atmosphere coupling and synoptic-scale persistence. In contrast,  $\kappa 

Figure 1. Spatial distributions of the inferred parameters of the fractional Hawkes process: (A) baseline rate  $\mu$ , (B) fractional memory order  $\alpha$ , and (C) excitation amplitude  $\kappa$ . Colored stars indicate *medoid stations*, defined as representative grid points selected from clustering of the parameter vectors  $(\mu, \kappa, \alpha)$ , which typify the characteristic hydroclimatic regimes (arid north, central transition zone, and humid south). High  $\mu$  corresponds to high exogenous drought hazard; small  $\alpha$  indicates long memory (slow power-law decay of  $\phi(t) \sim t^{-\alpha}$ ); and large  $\kappa$  reflects strong self-excitation and internal feedback.

## 4.2 Distribution and latitudinal dependence of fractional memory

The distribution of the fractional order  $\alpha$  across all locations (Fig. 2A) is right-skewed: most grid points fall in the range 0.3–0.5, with a secondary tail extending toward 0.7–0.8. This confirms that drought persistence is not spatially uniform but spans a continuum of memory regimes.

Panel B of Fig. 2 quantifies the latitudinal structure through a third-degree polynomial fit. A systematic increase of  $\alpha$  with latitude is evident: subtropical and centro-northern regions ( $\sim$ 20°-35°S) exhibit the lowest  $\alpha$  (0.2–0.4), corresponding to slow memory decay and long-lived persistence, whereas southern regions approach  $\alpha \sim$  0.6–0.7, indicative of faster relaxation. Central latitudes display elevated heterogeneity, with closely spaced locations showing contrasting  $\alpha$  values, suggesting a mosaic of persistence regimes shaped by local hydrology, orographic forcing, and circulation patterns. In physical terms, this meridional organization implies that the effective drought "memory horizon" tends to shorten poleward, consistent with stronger seasonal resetting and higher storm-track variability at higher latitudes.

Figure 2. (A) Histogram and kernel density estimate of the fractional memory order  $\alpha$  across all analyzed grid points. (B) Relationship between  $\alpha$  and latitude, with a third-degree polynomial fit (dashed line). The poleward increase in  $\alpha$  reflects a progressive reduction in drought memory: lower  $\alpha$  in the subtropical north corresponds to slow decay of the memory kernel  $\phi(t) \sim t^{-\alpha}$  (long persistence), whereas higher  $\alpha$  at higher latitudes corresponds to faster recovery.

#### 4.3 Temporal response and the role of tempering

To examine how finite-memory behavior emerges from the underlying fractional dynamics, Fig. 3 compares the modeled intensity response  $\lambda(t)$  and the corresponding memory kernels  $\phi(h)$  for a representative set of *medoid stations*. These medoids are

325

330

representative grid points identified from clustering of the parameter vectors  $(\mu, \kappa, \alpha)$ ; each medoid corresponds to the location whose local parameter triplet lies closest to the centroid of its regional cluster. They thus typify the dominant hydroclimatic behavior within each major regime—arid north, central transition zone, and humid south—providing a concise yet physically meaningful sampling of the full spatial variability.

Panel (a) shows the simulated tempered intensity  $\lambda(t)$  for all medoid stations under a common sequence of imposed exogenous forcing bursts (shaded intervals), using each station's fitted  $(\mu, \kappa, \alpha)$  and a fixed tempering rate  $\theta = 0.06$ . Although all sites experience the same external forcing, their relaxation trajectories diverge markedly. Stations with small  $\alpha$  and large  $\kappa$  exhibit slow post-burst decay and strong self-excitation, consistent with persistent feedbacks and long drought memory. Conversely, medoids with larger  $\alpha$  relax rapidly toward baseline, indicating weaker endogenous persistence and more efficient hydrological re-equilibration.

Panel (b) displays the associated memory kernels  $\phi(h)$  on a logarithmic scale. Dashed curves represent the non-tempered fractional kernels  $\phi(h) \propto h^{-\alpha}$ , which decay algebraically and imply infinite memory. Solid curves correspond to their exponentially tempered counterparts  $\phi(h) \propto h^{-\alpha}e^{-\theta h}$ , where the exponential factor truncates the long tail, introducing a finite persistence horizon  $\tau_m = 1/\theta$ . The contrast between dashed and solid curves highlights the physical role of tempering: it preserves power-law dependence at short and intermediate timescales while preventing unbounded accumulation of historical influence. In hydroclimatic terms, tempering represents finite-capacity memory mechanisms—such as seasonal hydrological resetting, soil-moisture depletion, or large-scale circulation regime shifts—that limit effective drought memory while retaining its fractional, scale-free character.

## 4.4 Effective finite-memory scale

To translate the inferred parameters into an interpretable persistence diagnostic, we derived an effective finite-memory timescale  $\tau_m = 1/\theta_{\rm eff}$  from the estimated  $(\kappa, \alpha)$  fields using Eq. (26) and the subcriticality condition  $\eta^* = 0.9$ . The resulting spatial distribution (Fig. 4) characterizes the duration over which past dry conditions continue to elevate the probability of new drought onsets.

Most of the domain exhibits relatively short  $\tau_m$ , implying that once external forcing weakens, the system relaxes toward baseline on comparatively short horizons. However, locally enhanced  $\tau_m$  appears in parts of the arid north and the far south, indicating regions where the effective damping is weak and drought memory decays more slowly. By contrast, central Chile shows shorter  $\tau_m$ , suggesting that despite relatively strong excitation ( $\kappa$ ), its dynamics re-equilibrate more quickly. These patterns confirm that finite climatic memory is not spatially uniform but emerges from the combined control of fractional persistence ( $\alpha$ ), excitation intensity ( $\kappa$ ), and local hydroclimatic setting.

# 4.5 Summary of spatial-temporal organization

Taken together, the spatial patterns of  $(\mu, \alpha, \kappa, \tau_m)$  reveal a coherent organization of drought onset dynamics across Chile. The baseline hazard  $\mu$  follows the climatological aridity gradient; the memory order  $\alpha$  isolates regions where past dry conditions exert prolonged influence; the excitation amplitude  $\kappa$  quantifies the internal reinforcement of dryness once initiated; and the

Figure 3. Comparative dynamics of medoid stations (representative locations selected to typify the main hydroclimatic regions) illustrating finite-memory effects in the tempered fractional Hawkes process (TFHP). (a) Modeled tempered intensity  $\lambda(t)$  for all medoid stations during a sequence of imposed exogenous bursts (shaded regions). Each colored curve corresponds to one medoid, simulated with its fitted  $(\mu, \kappa, \alpha)$  and a fixed tempering rate  $\theta = 0.06$ . Differences in recovery rate reflect variations in excitation strength  $\kappa$  and memory order  $\alpha$ . (b) Corresponding memory kernels  $\phi(h)$  on a logarithmic scale. Dashed curves show the non-tempered fractional kernels  $\phi(h) \propto h^{-\alpha}$  (infinite-memory limit), while solid curves show the exponentially tempered kernels  $\phi(h) \propto h^{-\alpha} e^{-\theta h}$ , which impose a finite persistence horizon  $\tau_m = 1/\theta$ . The comparison demonstrates how tempering stabilizes the process by truncating the algebraic tail, providing a finite yet still long-lived drought memory.

derived  $\tau_m$  translates these parameters into an effective finite memory horizon. These diagnostics demonstrate that the tempered fractional Hawkes framework captures both short-term clustering and long-range persistence of dry-spell onsets within a single physically interpretable structure. In particular, the combination of small  $\alpha$  and large  $\kappa$  emerges as a marker of dynamically fragile hydroclimatic regimes, in which droughts are both persistent and self-sustaining.

# 5 Discussion and Conclusions

# 5.1 Theoretical and methodological implications

The proposed *Tempered Fractional Hawkes Process* (TFHP) constitutes a consistent mathematical and physically interpretable framework for modelling the joint phenomena of persistence and clustering in hydroclimatic extremes. Unlike conventional Hawkes models with exponential memory kernels (Hawkes, 1971; Bacry et al., 2015), which impose a short-memory, Markovian relaxation, the fractional formulation embeds a power-law kernel  $\phi(t) \sim t^{-\alpha}$  derived via the Caputo fractional derivative. This structure captures algebraically decaying correlations and links stochastic point-process dynamics with fractional

Figure 4. Spatial distribution of the effective finite-memory scale  $\tau_m = 1/\theta_{\rm eff}$  derived from the tempered fractional Hawkes equivalent under  $\eta^* = 0.9$ . Larger  $\tau_m$  indicates longer drought memory and slower relaxation of dry conditions, whereas smaller  $\tau_m$  denotes faster recovery. The heterogeneous pattern reflects spatial differences in fractional persistence  $(\alpha)$  and excitation strength  $(\kappa)$ .

relaxation equations, which themselves find well-defined analogues in anomalous diffusion, visco-elasticity, and other non-Markovian systems (Hainaut, 2020; Chen et al., 2021; Chechkin and Metzler, 2017).

380

385

400

The introduction of an exponential tempering factor  $e^{-\theta t}$  regularises the kernel and ensures not only memory truncation (finite memory horizon) but also dynamic stability (ensuring subcriticality of excitation). This extension parallels the tempered fractional diffusion frameworks found in stochastic physics (Meerschaert and Sabzikar, 2015; Li et al., 2022) and resolves a key limitation of purely fractional models: their formal tendency toward unbounded accumulation of memory influence and potential non-stationary growth of the internal excitation budget. In the TFHP, the parameter set  $(\mu, \kappa, \alpha, \theta)$  defines a closed dynamical system:  $\alpha$  governs the rate of memory decay,  $\kappa$  quantifies self-excitation strength,  $\theta$  imposes the finite-memory horizon  $\tau_m = 1/\theta$ , and  $\mu$  regulates the background exogenous forcing. This yields a parsimonious yet physically grounded model that unifies long-memory relaxation and self-exciting point-process dynamics in a single formalism.

From a methodological vantage point, this study advances the state of the art in three principal respects. First, it provides an inference framework capable of estimating fractional memory parameters at regional scale using event-based precipitation sequences, thereby bridging point-process theory and hydro-climatological data analysis in a nonlinear systems context. Second, it introduces an effective tempering parameter  $\theta_{\rm eff}$ , derived via the subcriticality condition  $\eta^* 

410

430

offered by the TFHP of exogenous forcing ( $\mu$ ) and endogenous feedback ( $\kappa$ ) enables assessment of how anthropogenic or largescale climatic forcings modulate intrinsic system memory—providing a path toward physically informed drought predictability and resilience studies.

## 5.3 Future directions

There are several promising extensions of the present work that would deepen its theoretical and practical significance. First, the current analysis treats each grid cell as an independent univariate point process; a natural next step is the formulation of a multivariate or spatially coupled TFHP, capable of representing interactions among neighbouring sites and capturing the spatio-temporal propagation of drought clusters in a fractional–tempered Hawkes network framework (Ogata, 1988; Bacry et al., 2015). Second, the explicit inclusion of large-scale climate indices (e.g., ENSO, PDO, SAM) within the conditional intensity function,

$$\lambda(t) = \mu + \int_{0}^{t} \phi_{\alpha,\theta}(t-s) \, \mathrm{d}N(s) + \mathbf{b}^{\top} \mathbf{X}(t)$$

would allow quantification of the relative influence of external drivers versus internal persistence, thereby bridging stochastic process modelling and causal climate diagnostics. Third, the integration of TFHP-derived features into hybrid machine-learning or physics-informed learning systems could enable data-driven prediction of drought-onset probabilities while preserving interpretability—a direction aligned with the emerging interest in explainable Earth-system modelling. Finally, the formal equivalence between fractional Hawkes dynamics and tempered Caputo-type equations suggests a broader applicability beyond hydrology, including ecological regime shifts, wildfire occurrence, epidemic spread and other systems characterised by long-memory and self-excitation.

# 5.4 Concluding remarks

In summary, this study establishes a unified theoretical and empirical framework for characterising drought persistence in hydro-climatic systems via tempered fractional point processes. By embedding fractional memory and finite damping within the Hawkes structure, the TFHP captures both clustering and long-range dependence of dry-spell onsets in a physically interpretable manner. Applied to four decades of daily precipitation data over Chile, the model reveals coherent spatial patterns of climatic memory, excitation strength and recovery dynamics that align with known hydroclimatic regimes and large-scale circulation controls. Beyond its regional application, the TFHP contributes a general methodological and conceptual tool for diagnosing persistence in non-Markovian and nonlinear geophysical systems—thus bridging the conceptual gap between stochastic process theory and climate dynamics. In doing so, it provides not only a new modelling tool but also a formal language to quantify how memory, feedback and finite capacity jointly shape the resilience of the climate system.

Code and data availability. All analysis code (Python notebooks for onset extraction, parameter inference, and figure generation) is available at https://github.com/mauricio-herrera/tempered-fractional-hawkes-drought-persistence

The curated precipitation dataset (460 locations ERA5-based daily records, 1980–2022) is archived at Harvard Dataverse, DOI: https://doi.org/10.7910/DVN/IBZRJO. Raw ERA5 fields are available from the Copernicus Climate Data Store.

Author contributions. M. Herrera-Marín conceived the study, performed the analysis, interpreted the results, and wrote the manuscript.

Competing interests. The author declares no competing interests.

*Acknowledgements.* The author thanks colleagues for discussions on fractional dynamics and point-process inference. No specific funding was received for this research.

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
