# Peer review of "A tempered fractional Hawkes framework for finite-memory drought dynamics"

_EGUsphere, 2025_

## Referee Comment (RC1)

**Report on the article**
**A tempered fractional Hawkes framework for finite-memory drought dynamics**

The article introduces tempered and untempered fractional Hawks processes which are self-exiting point processes with decaying memory, and derives a mean field formulation of the intensity $\lambda(t)$. Given rainfall data from Chile, a log-likihood maximization is performed to fit the parameters of an untempered fractional Hawkes process to the occurrence of drought periods. Based on these parameters, the temperedness parameter $\theta$ is derived and the results are compared.

While tempered fractional calculus is established elsewhere [1], the tempered factional Hawkes process (TFHP) is to the best of my knowledge new, but the author is advised to compare their formulation to the work [2]. The introduction of the TFHP contains multiple mathematical inaccuracies and inconsistencies that are listed below. In particular, the interpretation of the parameter $\alpha$ seems reversed, thus making the interpretation of the results problematic. Additionally, the newly introduced TFHP is effectively not used in the numerical method, which almost exclusively relies on untempered fractional Hawkes processes.

Overall, these problems result in inconsistencies in both the theory and the application part of the article. The theoretical issues are certainly fixable, even though this would require rewriting most of Section 2. The problem that the TFHP is not actually used in the numerical method is a more fundamental weakness of the article. In view of the improvable methodology, the results do not seem to provide a very clear interpretation, in particular with the inconsistencies regarding the parameter $\alpha$ (see comment (I) below).

**Major comments.**

(I) The memory kernel of the fractional Hawkes process is introduced in (3) as

$$\phi_\alpha(t) = \frac{\kappa t^{\alpha-1}}{\Gamma(\alpha)}, \qquad 0 < \alpha < 1, \quad \kappa > 0. \tag{3}$$

The mean field-intensity is given in (4) by

$$\lambda(t) = \mu + \int_0^t \phi_\alpha(t - s)\lambda(s)\,\mathrm{d}s. \tag{4}$$

In line 116 it is stated that "*Small $\alpha$ correspond to extremely persistent excitations, i.e. slow-memory decay.*" The same interpretation of the parameter $\alpha$ is used throughout Section 4 to interpret the results. Inspecting the definition of $\phi_\alpha(t)$ in (3) shows that the memory decay like $t^{\alpha-1}$ such that small $\alpha$ correspond to quicker memory decay and $\alpha$ close to 1 correspond to slower memory decay. Hence, the role of $\alpha$ is reversed. Interestingly, in Section 4 it is mentioned on multiple occasions that $\phi(t) \sim t^{-\alpha}$ which would fit the interpretation of the parameter $\alpha$, but contradicts (3). If indeed the reversed meaning of the parameter $\alpha$ is used in Section 4 this would render any interpretations of the results void.

(II) The Caputo derivative $D_C^\alpha$ is never defined. This is relevant since it seems that the Caputo derivative is confused with the Riemann–Liouville derivative on multiple occasions, leading to incorrect derivations. For clarity, the Caputo derivative is commonly defined as

$$[D_C^\alpha f](t) = \frac{1}{\Gamma(1-\alpha)} \int_0^t (t-s)^{-\alpha} f'(s) \, ds.$$

On the other hand, the Riemann–Liouville derivative is commonly defined as

$$[D_{RL}^\alpha f](t) = \frac{1}{\Gamma(1-\alpha)} \frac{d}{dt} \int_0^t (t-s)^{-\alpha} f(s) \, ds.$$

These definitions, are similar, but differ by a term

$$[D_C^\alpha f](t) = [D_{RL}^\alpha f](t) - \frac{f(0)}{\Gamma(1-\alpha)} t^{-\alpha}.$$

In particular, the Caputo derivative is zero in constants while the Riemann–Liouville derivative is not.

Based on (8), it seems that the RL-derivative is used instead of the Caputo derivative. However, the Laplace transform forumala in line 140 is only valid for the Caputo derivative. So if indeed the Caputo derivative is used, then the term involving $\mu$ simply does not occur in the Caputo fractional differential equation (9). On the same note, the comment that as $\alpha \to 1^-$ the equation reduces to $\dot\lambda(t) = \kappa\lambda(t) + \mu$ is wrong. Indeed as $\alpha \to 1^-$, the Caputo derivative approches the classical derivative, but taking the derivative in (4) with $\phi_\alpha(t) = \kappa$ clearly yields $\dot\lambda(t) = \kappa\lambda(t)$.

Very similar confusions arise in Section 2.6 for the tempered Caputo derivative. In particular (18) is contradictory, since the first line is non-zero for constant functions $f$ while the second line is zero for constant $f$. In this case the incorrectness comes from the fact that the reference used [1] works with the Riemann–Liouville derivative, while the present article attempts to work in the Caputo framework. Either way, the definition is unclear and inconsistent.

Lastly, in line 173, it is stated that $D_{C,\theta}^\alpha$ converges to the classical first derivative plus an exponential damping term as $\alpha \to 1^-$. For the tempered Caputo derivative this convergence holds without additional damping terms, for all $\theta \geq 0$. The statement from line 173 may well be true for tempered RL derivatives.

(III) In Section 3 the computational method is presented. Given an observation of times $t_i$, the parameters $(\mu, \kappa, \alpha)$ of an *untempered* fractional Hawkes are fitted to the data. Only afterwards is the tempering parameter $\theta$ derived using the formula

$$\theta_{\mathrm{eff}} = \left(\frac{\kappa}{0.9}\right)^{1/\alpha}.$$

Hence, the TFHP, which has been introduced and named as the main motivation of the article, is not directly part of the computation. In line 198 it is mentioned that also

fitting the parameter $\theta$ is *numerically fragile.* It seems to me that finding a stable way to fit the parameter $\theta$ to the data, and thereby fitting a TFHP to the observations, is a crucial step in the application of the theoretical objects introduced.

Additionally, the implementation is only discussed briefly. The optimization of the log-likelihood is only mentioned in one sentence in line 192. A more detailed explanation of what is being computed would greatly increase the understandability and credibility of the results.

**Minor comments.**

(i) In line 153 it is mentioned that a pure power-law kernel has *infinite memory* since "*all past events, no matter how old, continue to influence $\lambda(t)$ with nonzero weight*". In contrast to this a tempered kernel is said to have *finite memory.* This is somewhat misleading since also an exponentially decaying kernel is nonzero for all times. The key difference is that the integrated endogeneity (branching ratio) is infinite for a power-law kernel and finite for any tempered kernel. However, this relation has been computed incorrectly in (23), since

$$\eta_{\text{frac}} = \int_0^\infty \phi_\alpha(t)\,\mathrm{d}t = \frac{\kappa}{\Gamma(\alpha)} \int_0^\infty t^{\alpha-1}\,\mathrm{d}t = \infty.$$

**References**

[1] F. Sabzikar, M. Meerschaert, and J. Chen. Tempered fractional calculus. *Journal of Computational Physics* 293 (2015): 14-28.

[2] N. Gupta, and A. Maheshwari. Tempered fractional Hawkes process and its generalizations. *Journal of Mathematical Analysis and Applications*(2025): 129996.